# Synergistic Effect of Doxorubicin and siRNA-Mediated Silencing of Mcl-1 Using Cationic Niosomes against 3D MCF-7 Spheroids

**DOI:** 10.3390/pharmaceutics13040550

**Published:** 2021-04-14

**Authors:** Supusson Pengnam, Samarwadee Plianwong, Prasopchai Patrojanasophon, Widchaya Radchatawedchakoon, Boon-ek Yingyongnarongkul, Praneet Opanasopit, Purin Charoensuksai

**Affiliations:** 1Pharmaceutical Development of Green Innovations Group (PDGIG), Faculty of Pharmacy, Silpakorn University, Nakhon Pathom 73000, Thailand; supussonpengnam@gmail.com (S.P.); patrojanasophon_p@su.ac.th (P.P.); opanasopit_p@su.ac.th (P.O.); 2Pharmaceutical Innovations of Natural Products Unit (PhInNat), Faculty of Pharmaceutical Sciences, Burapha University, Saen Suk 20131, Thailand; samarwadee.pl@go.buu.ac.th; 3Creative Chemistry and Innovation Research Unit, Department of Chemistry and Center of Excellence for Innovation in Chemistry (PERCH-CIC), Faculty of Science, Mahasarakham University, Mahasarakham 44150, Thailand; widchaya.r@msu.ac.th; 4Department of Chemistry and Center of Excellence for Innovation in Chemistry, Faculty of Science, Ramkhamhaeng University, Bangkok 10240, Thailand; boonek@ru.ac.th; 5Bioactives from Natural Resources Research Collaboration for Excellence in Pharmaceutical Sciences, Faculty of Pharmacy, Silpakorn University, Nakhon Pathom 73000, Thailand

**Keywords:** siRNA, Mcl-1, cationic niosomes, combination chemotherapy, breast cancer

## Abstract

Chemotherapy is a vital option for cancer treatment; however, its therapeutic outcomes are limited by dose-dependent toxicity and the occurrence of chemoresistance. siRNAs have emerged as an attractive therapeutic option enabling specific interference with target genes. Combination therapy using chemotherapeutic agents along with gene therapy could be a potential strategy for cancer management, which not only improves therapeutic efficacy but also decreases untoward effects from dose reduction. In this study, a cationic niosome containing plier-like cationic lipid B was used to convey siRNA against anti-apoptotic mRNA into MCF-7 and MDA-MB-231 cells. Mcl-1 silencing markedly decreased the viability of MCF-7 cells and triggered apoptosis. Moreover, computer modeling suggested that the combination of doxorubicin (Dox) and Mcl-1 siRNA exhibited a synergistic relationship and enabled a dose reduction of each agent at 1.71 and 3.91 folds, respectively, to reach a 90% inhibitory effect when compared to single-agent treatments. Synergistic antitumor activity was further verified in a 3D spheroid culture which revealed, in contrast to single-agent treatment, the combination markedly decreased spheroid volume over time. Together, the combination therapy between Mcl-1 silencing and Dox exhibits a synergistic effect that may be exploited for novel breast cancer treatment.

## 1. Introduction

Adjuvant and neoadjuvant chemotherapy are currently standard treatment options for many stages of breast cancer [1,2]. Nevertheless, chemotherapy is often associated with adverse drug reactions, including bone marrow suppression, hair loss, and gastrointestinal irritation [3]. One of the most widely used chemotherapeutic agents for breast cancer is doxorubicin (Dox). Current chemotherapy often uses a combination of several chemotherapeutic agents (polychemotherapy), which provide attractive benefits, including increased efficacy and a decreased dose of each individual agent to reduce chemoresistance [4]. For breast cancer, Dox is often combined with other agents such as cyclophosphamide, 5-fluorouracil, paclitaxel, docetaxel, and mitomycin. Despite the advantages of drug combinations in chemotherapy, they are still associated with adverse side effects such as neutropenia and thrombocytopenia, which are observed in 74 and 25% of patients treated with Dox and a mitomycin combination, respectively [4]. Moreover, using Dox at a high dose is associated with cardiotoxicity [5]. Evidence supported that the incidence of cardiotoxicity increases along with increasing doses. The use of Dox at 500–550, 551–600, and over 600 mg/m^2^ relatively increases the cardiotoxicity incidence at 4, 18, and 36%, respectively [6,7]. These limitations likely result from the lack of tumor specificity for current chemotherapeutic agents. Currently, breast cancer is leading the most cancer found in female patients and was the second cause of death in the U.S. in 2020 [8]. Therefore, developments of novel therapeutic strategies for breast cancer are urgently needed to improve therapeutic outcomes and patients’ quality of life.

siRNA are short double-stranded RNA segments, usually composed of 21–23 nucleotides. siRNA is specifically able to bind to mRNA with a complemented nucleotide sequence, target it for degradation, and down-regulate the target protein [9,10]. Thus, siRNA has become of particular interest as a potential therapeutic option for genetically related diseases, including cancers [11]. With the ability to specifically interfere with the function of target genes, siRNA-based therapy is expected to have fewer off-target associated side effects. Moreover, siRNA can be used to down-regulate genes responsible for chemotherapy resistance and re-sensitize tumors to chemotherapy [12]. Apoptosis or programmed cell death is a fundamental biological process regulating homeostasis. Deregulation of this pathway is important in the pathology of many diseases as well as cancer [13]. As early as the 1970s, Kerr et al. proposed that hypergenesis sometimes results from the loss of apoptosis rather than an increase in mitosis [14]. The evasion of apoptosis is known as one of the hallmarks of cancer [15]. The initiation of apoptosis is governed by the balance of pro and anti-apoptotic proteins. The anti-apoptotic proteins, i.e., the Bcl-2 protein family, includes Bcl2-A1, Bcl-2, Bcl-xL, Bcl-w, and Mcl-1, which are pro-survival mediators preventing apoptosis [16,17]. The upregulation and hyperactivation of the Bcl-2 family have been described in cancers, which contributes to the survival of cancerous cells and the resistance to chemotherapy [16,18]. Bcl-2 is overexpressed in half of the human malignancies and 50–70% of breast cancer patients [19,20,21]. Mcl-1 is also a vital protein regulating breast cancer cell survival [22]. Overexpression of Mcl-1 is manifested in various human malignancies, including breast cancer which contributes to cell survival and conventional therapeutic resistance [16,23,24]. siRNA-mediated downregulation of Mcl-1 confers beneficial therapeutic effects toward breast cancer management. The siRNA against Mcl-1 suppressed the cell viability of MDA-MB-435 cells [25]. Moreover, Mcl-1 silencing in Dox-resistant MDA-MB-435 cells overexpressing P-gp, BCRP, Mcl-1, and survivin led to the loss of cell viability, suggesting that targeting Mcl-1 could serve as an alternative treatment for Dox-resistant cancer [25]. Additionally, transient transfection of miR-193b targeted and downregulated Mcl-1 mRNA and protein expression and noticeably increased Dox sensitivity in Dox-resistant MCF-7 cells [26]. The initiation of the apoptosis pathway eventually converges at the activation of the caspase cascade. However, the function of caspases can be blocked by a family of proteins through caspases binding termed inhibitor of apoptosis proteins (IAPs), e.g., NIAP, c-IAP1, XIAP, and survivin [27,28]. Among these IAPs, survivin exhibits significant overexpression in many tumors, especially in breast and lung cancer, but it is present at low levels or completely absent in healthy cells and tissues [29,30,31,32]. Moreover, targeting survivin with siRNA or miRNA re-sensitizes drug-resistant breast cancer cell lines to chemotherapeutic agents, including paclitaxel, taxol, and Dox [33,34].

Despite the attractive attributes of siRNA in cancer precision therapy, the success of siRNA-based therapy relies on safe and effective siRNA delivery systems. Current siRNA delivery systems can be classified into two categories, i.e., viral and non-viral vectors. Although viral vectors provide high transfection efficiency, they also possess several drawbacks, for instance, immunogenicity and low loading capacity. Therefore, non-viral vectors have emerged as alternative delivery systems with minimal safety concerns, which are broadly used for gene delivery. The advantages of non-viral nanoparticles include the simple production process and ease of structural modification [35]. Current non-viral gene delivery strategies include polymers, cell-penetrating peptides, and cationic lipids. Among these, lipid-based formulations such as liposomes are probably the ones most frequently used for human gene therapy in clinical trials [35,36]. Generally, the core component of liposomes is naturally derived phospholipids such as egg phosphatidylethanolamine to form bilayer membrane vesicles. Niosomes, a new generation gene delivery system with a core component of nonionic surfactants, are prepared through similar approaches used in the production of liposomes. They have emerged as safe and viable delivery alternatives to the liposomes with key attractive features, including biodegradability, biocompatibility, and less immunogenicity, as evidenced by numerous preclinical and clinical researches [37,38]. Moreover, nonionic surfactants possess higher chemical stability than phospholipids which can be degraded by oxidation or phospholipases [37]. The use of synthetic surfactants is generally more economical and reduces batch-to-batch differences associated with the usage of lipids derived from natural sources [39,40]. Cationic lipids are often combined in lipid vesicles, offering the adjustable electrostatic interaction with the cellular membrane and genetic materials to facilitate gene delivery through the selection of cationic lipids with various structures. However, there are only a few studies investigating the application of cationic niosomes for gene delivery [38].

The authors previously reported a novel cationic lipid that can be used to prepare cationic niosomes containing plier-like cationic lipid B (PCN-B). It can be used effectively for the delivery of siRNA with high transfection efficiency and minimal cytotoxicity in a green fluorescent screening model [41,42]. The anticancer effect of siRNA against anti-apoptotic genes, namely, Mcl-1, Bcl-2, and survivin, were first screened in breast cancer cell line MCF-7 and MDA-MB-231. Dox, a well-established chemotherapeutic agent routinely used to manage breast cancer, was selected for combination treatment. Here, this study provided a systematic approach to identifying the synergistic effect of Dox and a potent siRNA delivered by cationic niosomes, hypothesized as a novel therapeutic strategy for breast cancer treatment. Moreover, there is still a lack of empirical algorithm study in the synergistic effect of combining chemotherapeutic agent and siRNA. One of the obstacles in cancer therapy is the mathematical translation of enormous information into therapeutic application since the interpretation of acquiring knowledge is time-consuming and high cost [43]. CompuSyn, based on mass-action law providing global agreement and standardization of synergy definition, was used to analyze synergistic activity and dose prediction for Dox and candidate siRNA targeting apoptosis in combination therapy. Finally, the anticancer effect of Dox and siRNA combination therapy was verified against a breast cancer cell line cultured as a 3D spheroid.

## 2. Results

### 2.1. Characterization of Cationic Niosomes

The morphological characteristic of cationic niosomes and nioplexes were observed via transmission electron microscope (TEM). PCNs had a unilamellar spherical shape and smooth border. The particles obtained from the TEM image were approximately 87 to 186 nm for PCN-B and PCN-B/siRNA complexes, respectively. While the particle size of PCN-B and PCN-N/siRNA complexes from Zetasizer nano ZS were 141.3 ± 7.5 and 697.6 ± 50.9 nm, respectively (Figure 1a,b). The particles obtained from the TEM image were in the dried state and consequently smaller particle sizes, while the bigger particles obtained from Zetasizer nano ZS were in the water-containing state. The zeta potential of PCN-B and PCN-B/siRNA complexes were 54.8 ± 0.2 and 10.4 ± 5.4 mV, respectively.

### 2.2. Cellular Uptakes and Intracellular Distribution of siRNA Complexes by Flow Cytometry and Confocal Laser Scanning Microscopy

First, cellular uptake of fluorescently-labeled siRNA with Alexa Fluor 488 (siAF488) was performed to ensure that PCN-B/siRNA complexes could effectively deliver siRNA into MCF-7 and MDA-MB-231 cells. The uptake of siAF488 can be evaluated by measuring the green fluorescent signal of siAF488 through flow cytometry analysis. The ratio of transfection agents used in this study was guided by the reported weight ratios of transfection agents: siRNA presenting high silencing efficiency, i.e., 2.5 and 15 for lipofectamine^®^ 2000 (Lipo2k) and PCN-B, respectively [42]. Transfection was performed under the presence of 10% serum. The result revealed high cellular uptake in both cell lines (Figure 2a,b). The transfection efficiency of siRNA complexes of PCN-B and Lipo2k was as high as 95 and 84%, respectively, in MCF-7 cells. As presented in Figure 2, the mean fluorescent intensity (MFI) of cells transfected with siAF488 using PCN-B as transfection reagent was significantly higher than that of Lipo2k, suggesting that PCN-B could deliver high amounts of siRNA into both cell lines. This result suggested that PCN-B might be a potential cationic lipid with a high siRNA delivering capacity for breast cancer cells.

In line with this observation, confocal laser scanning microscope (CLSM) also revealed that PCN-B could deliver siAF488 into the desired cells (Figure 3). Mostly, the cells taken up the complexes via the endocytosis mechanism presented highly in the cytoplasm. Cells transfected with PCN-B manifested high cellular accumulation of siAF488 in both cell lines is consistent with flow cytometry data.

### 2.3. The Effect of Anti-Apoptosis siRNA Delivery on the Viability of Breast Cancer Cells

The viability of the breast cancer cell line MCF-7 and MDA-MB-231 upon exposure of siRNA targeting three anti-apoptotic genes, namely, Mcl-1, Bcl-2, and survivin, was examined by MTT assay. The results revealed that Bcl-2 and survivin silencing did not affect the viability of both cell lines (Figure 4). However, Mcl-1 silencing noticeably decreased the viability of MCF-7 by approximately 30% (Figure 4a), while such an effect was not observed in MDA-MB-231 (Figure 4b), suggesting that MCF-7 cells were especially sensitive to Mcl-1 downregulation and identified Mcl-1 as a potential target for MCF-7 cells. Therefore, Mcl-1 silencing was selected for further experiments, which hypothesized that combining Mcl-1 siRNA and Dox might reduce the dose of each individual agent for the inhibition of MCF-7 cells.

### 2.4. Combination of Dox and Mcl-1 siRNA with a Non-Constant Concentration Ratio

The driving hypothesis is that a combination therapy between the chemotherapeutic agent and the siRNA-targeting essential gene for cancer survival could yield the desired synergistic pharmacological effects. First, the combination of Dox and Mcl-1 siRNA was examined in MCF-7 cells to observe the cell viability by MTT assay (Figure 5a). The ratios of Dox:Mcl-1 siRNA were increased from 0.2 to 9.2 using the fixed concentration of Mcl-1 siRNA (0.1 µM) to investigate which ratio of Dox:Mcl-1 siRNA exhibited the synergistic effect. The combination of Dox and non-targeted siRNA (siNT) was also carried out as a mock control at those same ratios. The inhibition effect of the Dox:Mcl-1 siRNA combination against MCF-7 cells remained unchanged at ratios ranging from 0.2–0.9. However, the inhibitory effect of the combination markedly increased as the ratio rose above 0.9. The combination at a ratio between 2 and 4 could reduce the cell viability about 1.7 times when compared to a single treatment of Dox at the same concentration.

The data was also interpreted by CompuSyn software to assess the combination index (CI), which reflected synergistic outcomes as follows: CI < 1, CI = 1 and CI > 1, indicating synergistic, additive, and antagonistic effects, respectively. The synergistic effect of Dox:Mcl-1 siRNA was observed at ratios above 2, which presented IC < 1 as shown in Figure 5b [43,44,45]. The ratio of Dox:Mcl-1 siRNA above 2 was in favor of a dose reduction of both agents (Figure 5b). Herein, the ratio of Dox:Mcl-1 siRNA at 2.5 was selected for further experiments.

### 2.5. Combination of Dox and Mcl-1 siRNA with a Constant Concentration Ratio

The synergistic effect of Dox and Mcl-1 siRNA was further investigated with the constant ratio experiment. In this experiment, cells were treated with five different concentrations of Dox and Mcl-1 siRNA. While the ratio of Dox:Mcl-1 siRNA was fixed at 2.5 for all conditions, the concentration of Dox and Mcl-1 siRNA in this analysis ranged from 0.25–0.75 and 0.1–0.3 µM, respectively, resulting in 1-, 1.5-, 2-, 2.5-, and 3-fold increases in the concentration of the combination compared to the lowest concentration baseline (Figure 6a). The cell viability was evaluated by MTT assay and further analyzed by CompuSyn software. For control, the viability of cells treated with a single agent at the same concentration as that used in the combination was also examined and presented as dashed lines (Figure 6a). Although some cytotoxic effects were observed in the mock siRNA control (2.5Dox:siNT), the synergistic effect of 2.5Dox:Mcl-1 siRNA was detected at every concentration point tested. Of note, the cytotoxic activity of Mcl-1 siRNA remained unchanged within the concentration range tested (0.1 to 0.3 µM) (Figure 6a). Cell density and morphology were also observed under an inverted microscope (Figure 6b). The density of cells treated with the combination markedly decreased compared to cells individually treated with each agent at the same concentration. The loss of cell density and the presence of shrunk and detached cells increased by increasing the concentration of the combination.

The synergistic effect of the constant ratio of the concentration was further analyzed by CompuSyn software, which generates computerized simulation data from various doses and fraction of effect (Fa) [43,46,47]. The simulated data was simplified as a combination index plot (CI-Fa) and a dose reduction index plot (DRI-Fa) for a holistic view of the synergistic level and a number-fold dose reduction, respectively, at various Fa (Figure 6c,b) [43,46,47]. The result indicated that at this ratio, Dox and Mcl-1 siRNA showed synergy and a favorable dose reduction on a broad range of effects. The combination’s predicted dose achieved IC50, IC75, and IC90 (Fa 0.5, 0.75, and 0.9, respectively), while Dox and Mcl-1 siRNA exhibited synergistic effects (CI = 0.60, 0.67, and 0.84, respectively), enabling dose reductions of each agent by 2.5–4-fold. As the dose needed to reach the high inhibitory effect is considered more relevant in clinical settings, IC90 was thereby selected for further testing using the 3D spheroid model. The simulation also suggested that the combination at Fa 0.9 showed a favorable dose reduction of Dox and Mcl-1 siRNA at 1.71 and 3.91, respectively.

### 2.6. mRNA Expression

The Mcl-1 mRNA expression level of the MCF-7 cells treated 0.1 µM of Mcl-1 mRNA, using PCN-B or Lipo2K as the transfection agent with and without 0.25 µM of Dox in combination, was determined by real-time PCR. The siNT complex was employed as a negative control for each formulation. The Mcl-1 mRNA expression level was normalized with untreated cells and presented as % relative mRNA expression. The Mcl-1 expression level was not affected by the Dox or siNT treatment transfected with either PCN-B or Lipo2K (Figure 7: lane 2, 3, 4, 7, 8). Approximately a 60% reduction of Mcl-1 mRNA was observed in the Mcl-1 siRNA treatment with either Lipo 2K (Figure 7: lane 5, 6) or PCN-B (Figure 7: lane 9, 10). The addition of Dox did not affect the Mcl-1 downregulation efficiency of the Mcl-1 siRNA treatment (Figure 7: lane 6, 10).

### 2.7. Detection of Apoptosis by Hoechst 33342 and SYTOX Green Double Staining Method

Our analysis of Mcl-1 siRNA revealed that it can decrease the viability of MCF-7 cells. To further investigate if the decrease in cell viability was mediated through induction of apoptosis, MCF-7 cells treated with Mcl-1 siRNA alone (0.1 µM) or in combination with Dox were analyzed for condensed chromatin, a hallmark of cells undergoing apoptosis, using Hoechst 33342 and SYTOX Green double staining technique [48]. In this assay, Hoechst 33342 was used as a cell-permeable DNA labeling dye, which stained apoptotic cells more brightly than normal cells. The green fluorescence dye SYTOX™ Green, which cannot cross intact cell membranes, was used to stain dead cells. The stained MCF-7 cells were observed under white light and an inverted fluorescence microscope (Figure 8). A single treatment of Dox, which had been reported to induce apoptosis in MCF-7 cells, was also incorporated as a positive control [49].

As seen in the fluorescence images, the morphology of untreated and siNT treated cells exhibited a smooth-spheroid nucleus with normal chromatin distribution as evidenced by uniformly bright blue fluorescence and a lack of green fluorescent dead cells (Figure 8a,b). In contrast, the dominant morphology of the apoptotic cells was observed in the cells treated with a high concentration of Dox (0.75 µM), Mcl-1 siRNA, or Dox:Mcl-1 siRNA. The cell density markedly decreased when compared to untreated cells under bright-field microscopy. The volume of the nucleus and cytoplasm also decreased due to cell shrinkage from apoptosis. Moreover, chromatin condensation significantly increased when compared to untreated cells, as indicated by cells exhibiting the bright blue fluorescence of apoptotic chromatin. Green fluorescence was found in both the Dox and Mcl-1 siRNA treatment which might be the consequence of cell membrane rupture in late apoptosis or necrosis. The percentage of apoptotic and dead cells was significantly higher in the 2.5Dox:Mcl-1 siRNA treated group when compared with the Mcl-1 siRNA or the Dox single treatment. Noticeably, more dead cells were found in the Dox than in the Mcl-1 siRNA group. These findings indicated that combining Dox with Mcl-1 siRNA increases the induction of apoptosis in MCF-7 cells.

### 2.8. MCF-7 3D Spheroid Model

In this experiment, the anticancer activity of 2.5 Dox:Mcl-1 siRNA (1.25 µM Dox and 0.5 µM Mcl-1 siRNA, IC_90_) was examined with a MCF-7 3D spheroid culture. The 3D spheroids were imaged every 2 days for 8 days to evaluate the change in spheroid morphology and size (Figure 9). The necrotic core of spheroids was detected on day 4 except in the spheroid treated with the combination. The untreated spheroid clearly showed that the necrotic core continuously expanded until day 8. The spheroid treated with Dox or the Mcl-1 siRNA single agent was slightly smaller than the untreated spheroid. However, the spheroid size remained unchanged or only slightly increased under these conditions, suggesting that single treatments were insufficient to stop the growth of tumor spheroids. It was noticeable that the Dox-treated spheroid appeared more diffused at day 6 onward. On the contrary, the combination of Dox:Mcl-1 siRNA stunted the growth of the spheroid at an early time point and markedly shrunk the tumor spheroid at a later time point, indicating that the Dox:Mcl-1 siRNA combination treatment was more effective in breast tumorsphere inhibition than the single treatment of each individual agent.

## 3. Discussion

siRNA-based therapy has become an attractive therapeutic strategy due to its ability to interfere with desired mRNA targets specifically. The anticancer activity of siRNA targeting anti-apoptotic genes, including Mcl-1, Bcl-2, and survivin, without chemotherapeutic drug treatment, were first screened against breast cancer cell lines to identify attractive genes which may be combined with existing chemotherapeutic agents to increase anticancer effects. The authors’ results suggested that the Bcl-2 and survivin siRNA treatment was not effective at inducing cell death in the absence of the chemotherapeutic agent in both the MCF-7 and MDA-MB-231 cells. This finding is in line with several other reports. Firstly, Beh et al. demonstrated that Bcl-2 siRNA downregulated Bcl-2 mRNA by over 70% but did not affect the viability of MDA-MB-231 cells [50]. This could be explained by the relatively long half-life of the Bcl-2 protein. Additionally, Talaiezadeh et al. investigated the effect of Bcl-2 gene silencing in MCF-7 cells. They reported a decrease of more than 90% in Bcl-2 mRNA on the siRNA treatment; though, apoptotic cell death remained unchanged. The researchers suggested that the Bcl-2 targeting might trigger cell death through other pathways such as autophagy. Indeed, there is evidence indicating that the Bcl-2 protein can interact with Beclin 1 [51,52]. Bcl-2 reduction results in the release of Beclin1 and activation of autophagy. Moreover, autophagy can interrupt the apoptosis pathway by breaking down intracellular structures such as Golgi apparatus and mitochondria, an organelle important for the intrinsic apoptosis pathway supplying energy and nutrients needed for cell survival [53,54]. Trabulo et al. investigated the effect of survivin siRNA using Lipo2k as a delivery system in three cell lines, including A549, HeLa, and MCF-7. Although the cell viability of A549 and HeLa was reduced after the siRNA targeting survivin treatment, it did not significantly affect the viability of the MCF-7 cells [55].

Interestingly, the authors’ results demonstrated that Mcl-1 targeting effectively decreases the viability of MCF-7 cells, although such an effect was not detected in MDA-MB-231 cells. This was consistent with a previous study by Hamidrez et al., which reported that the Mcl-1 siRNA delivery did not decrease cell viability in MDA-MB-231 [25]. Mcl-1 is an anti-apoptotic protein in the Bcl-2 family which plays an important role in apoptosis induction. Mcl-1 is involved in many pathways, including the phosphatidylinositol-3 kinase (PI3K) pathway, the mTOR pathway, and the mitogen-activated protein kinase (MAPK) pathway. For these reasons, a significant increase of Mcl-1 may contribute to apoptotic escape and promote cancer cell survival via oncogenic signaling pathways [16]. Moreover, the Mcl-1 protein has a high turnover rate and a short half-life (i.e., 2–3 h) which is different from other Bcl-2 family proteins. Therefore, the pronounced effect of the Mcl-1 targeting detected in the authors’ study could be attributable to a rapidly changing Mcl-1 protein level [23,56]. Importantly, combining multiple targets and selecting the silencing target gene should be carefully considered to avoid therapeutic failure due to the inability of RNAi technology to eliminate pre-existing proteins that can obstruct the maximal effect of siRNA therapy [10,56].

The present study of a range of Dox and Mcl-1 siRNA combination ratios (0.2–9.2) indicated that Dox and Mcl-1 siRNA were synergistic with a CI value < 1 at 2–9.2 µM and suppressed the viability of MCF-7 cells by 50–90%. High inhibition is more relevant in clinical therapy for anticancer and antiviral agents, particularly those within the range of 75–90% [46,57]. The therapeutic outcome of drug combinations depends on the ratio and concentration of each agent. As the cardiotoxicity of Dox was frequently detected at a higher dosage, this toxicity was expected to decline with the lower dosage of Dox.

Normally, solid tumors consist of a central necrotic core and a metabolically active surface. A two-dimensional (2D) cell culture still has limitations due to the inability to recapitulate tumor microenvironments, including hypoxia, high interstitial fluid pressure, and low pH, as well as barriers for drug diffusion. Therefore, the antitumor activity of the combination of Dox and Mcl-1 siRNA was evaluated in the MCF-7 multicellular tumor spheroid model to simulate the tumor environment in vivo better while circumventing the usage of laboratory animals [58,59]. The MCF-7 cell spheroid was more susceptible to the combination of Dox:Mcl-1 siRNA than a single treatment at the same concentration. Mcl-1 siRNA might contribute to intracellular transduction inhibition and cell death induction via activation of the apoptosis pathway. Indeed, Mcl-1 overexpression was related to poor prognosis in many cancer types [60]. More importantly, Mcl-1 upregulation was associated with Dox resistance in breast cancer [25,61]. Long et al. successfully studied sensitized Dox-resistant MCF-7 cells to Dox with Mcl-1 downregulation using exogenous miR-193b [26]. Mcl-1 not only functions as a negative regulator of the pro-apoptotic protein responsible for mitochondrial/caspase activation but also binds with the BH3-only protein that is important for the protection of cytochrome C release from mitochondria. In addition, drug-induced apoptosis is antagonized by Mcl-1. Accordingly, it was concluded that Mcl-1 is an early indicator of survival and of apoptosis balance [62,63,64,65].

Dox can trigger apoptosis via activation of several pathways, including topoisomerase II inhibition, generation of free radicals, upregulation of Bax, caspase-8 and caspase-3, and downregulation of Bcl-2 protein expression [49,64]. It has been reported that there was a decrease in nuclear factor kappa-B (NF-κB) sensitized apoptosis in various cancer cells. Dox could decrease the NF-κB gene in MCF-7 cells [49]. This finding supported that Dox could induce apoptosis independently of the apoptotic cascades governed by Mcl-1 in MCF-7 cells. The combination therapies could trigger both Mcl-1 dependent and independent pathways, which might advocate the potential utility for cancer therapy [64]. Successful chemotherapy management depends on a balance between therapeutic response and side effects. A higher dose of Dox promoted an antitumor outcome; however, it was also associated with cardiotoxicity. Astonishingly, the combination in this study not only significantly increased the therapeutic effect but also reduced the dose of individual drugs, which was efficacious to decrease the side effects of Dox. Therefore, the combination of Dox and Mcl-1 siRNA served as a powerful antitumor regime as it targeted multiple vital pathways for cancer and thus warrants further assessment including in vivo experiments.

This study revealed an empirical strategy of combination therapy using a chemotherapeutic agent and the Mcl-1 siRNA therapy using CompuSyn software, which aided in the interpretation of the synergistic effect and the dose reduction in the drug combination. The authors’ simulation also suggested that the combination enabled a dose reduction of Dox and Mcl-1 siRNA at 1.71 and 3.91, respectively, at the predicted 90% inhibition response. Moreover, while single treatment of Dox or Mcl-1 siRNA alone was insufficient to slow down the expansion of the MCF-7 tumorsphere, the combination effectively prevented tumorsphere growth at early time points and decreased the tumorsphere size over time. In conclusion, this study served as further proof of concept that a chemotherapeutic agent-siRNA combination therapy could yield a superior response compared to that of single-agent treatments and highlights the combination treatment of Dox and Mcl-1 siRNA as an attractive candidate for novel breast cancer therapy.

## 4. Materials and Methods

### 4.1. Materials

Dox HCl and Span20 were purchased from Sigma Aldrich^®^, St. Louis, MO, USA. The synthesis of plier-like cationic lipid B (PCL-B) was kindly assisted by Ramkhamhaeng University and Mahasarakham University, Thailand. Cholesterol (Chol) was acquired from Carlo Erba Reagent (Milan, Italy). siAF488 was obtained from Qiagen (Santa Clarita, CA, USA). siRNA targeting apoptosis-related proteins including Mcl-1, Bcl-2, and survivin was obtained from Ambion™ Silencer™ siRNA, Thermo Fisher Scientific (Waltham, MA, USA). Lipo2k and siNT were purchased from Invitrogen (Carlsbad, CA, USA).

### 4.2. MCF-7 and MDA-MB-231 Cell Cultures

MCF-7 and MDA-MB-231 were cultured in Dulbecco’s Modified Eagle’s Medium (DMEM, Gibco, San Diego, CA, USA) and supplemented with 10% fetal bovine serum (Gibco) and 1% penicillin-streptomycin (Gibco) and then incubated in controlled 37 °C and 5% CO_2_ incubators. Cells were trypsinized and passaged when 70–80% confluency was reached.

### 4.3. Cationic Niosome Preparation

Cationic niosomes were formulated by the thin-film hydration method [66]. Niosome solution containing Span20, Chol, and PCL-B was prepared in a methanol:chloroform mixture (1:2%*v*/*v*). The concentrations of Span 20, Chol, and PCL-B in the niosome solution were 2.5, 2.5, and 2 mM, respectively. Thin-film was prepared by solvent evaporation under N_2_ gas and further hydrated with 1 mL of Tris-buffer (20 mM Tris and 150 mM NaCl, pH 7.4) at 60 °C. The resulting solution was sonicated for 30 min 2 cycles at 4 °C to reduce particle size. Afterward, visible particulate matter was eliminated by centrifugation at 15,000 rpm for 15 min. The cationic niosomes and nioplexes were evaluated by Zetasizer Nano ZS (Malvern Panalytical, Malvern, U.K.). The morphology of cationic niosomes and nioplexes were analyzed using a TEM (Philip Model TECNAI 20). Before the observation at an accelerating voltage of 80 kV, niosomes and nioplexes were dried on formvar-carbon film and stained with 1% uranyl acetate.

### 4.4. siRNA Cellular Uptake Assessment

The cellular uptake of siAF488, delivered by cationic niosome or Lipo2K control, was assessed through the flow cytometric method as previously described [67]. Briefly, cells were seeded at the concentration of 3.5 × 10^4^ cells/well in a 24-well plate. siAF488 complexes containing 30 pmol/well of siAF488 were then added to each well and incubated for 24 h. Afterward, culture media containing transfection reagents were discarded. Cells were then washed with phosphate-buffered saline (PBS), harvested by trypsinization, and fixed with 4% formaldehyde (Carlo Erba, Barcelona, Spain). The fluorescence signal was analyzed by a BD FACSCanto™ Flow Cytometer. The percentage of cellular uptake was defined as the percentage of transfected cells to non-transfected cells. MFI reflected the extent of siAF488 within transfected cells.

### 4.5. Cellular Imaging by Confocal Laser Scanning Microscope (CLSM)

Coverslips were placed inside a 24-well plate under sterile conditions before seeding the cells at 3.5 × 10^4^ cells/well. Complete media were maintained at 500 µL for every well. Cells were transfected with siAF488 complexes (30 pmol/well of siAF488) and incubated for 24 h. Afterward, the wells containing coverslip were gently washed with PBS and stained with staining solution containing 5 µg/mL of wheat germ agglutinin tetramethylrhodamine conjugate (WGA-TC) and 5 µg/mL of Hoechst 33342 trihydrochloride (Invitrogen, Carlsbad, CA, USA) for 15 min. Cells were then rinsed twice with PBS and fixed with 4% formaldehyde for another 15 min. The glass coverslips were mounted over glass slides with ProLong™ Diamond Antifade Mountant (Invitrogen, Carlsbad, CA, USA). The stained coverslips were loaded in an FV10i confocal laser scanning microscope, Olympus, to obtain a 60× oil immersion cell image.

### 4.6. siRNA Transfection

Transfection agents, including PCN-B or Lipo2k, were diluted in Tris-buffer. The anti-apoptotic siRNA (Mcl-1, Bcl-2, and survivin at 0.1 µM/well) was mixed with transfection agents for 30 min prior to transfection. The weight ratios of PCN-B/siRNA and Lipo2k/siRNA were fixed at 15 and 2.5, respectively. For combination experiments (2D and 3D), cells were incubated with media containing siRNA complex for 6 h prior to Dox treatment.

### 4.7. Evaluation of Cell Viability by the MTT Assay

Cells were seeded on a 96-well plate at 1 × 10^4^ cells/well and incubated overnight before transfection as described. The cells were treated with designated treatments for 24 h and then washed. Media were replaced with a fresh completed medium and incubated for another 48 h. Cell viability determination by MTT assay was examined at 72 h after transfections. Briefly, the cells were then incubated with 1 mg/mL of 3-(4,5-Dimethylthiazol-2-yl)-2,5-diphenyltetrazolium bromide (MTT) for 3 h. The optical density (OD) of crystal formazan dissolved in Dimethyl sulfoxide (DMSO) was achieved using VICTOR Nivo^®^ Multimode Microplate Reader, PerkinElmer at 550 nm. The percentage of viability was calculated by normalizing the OD of treated cells with untreated cells.
(1)$$\%\;\mathrm{cell}\;\mathrm{viability}=\frac{\left({\mathrm{OD}}_{\mathrm{treated}\;\mathrm{cells}}-{\mathrm{OD}}_{\mathrm{blank}\;\mathrm{DMSO}}\right)\times 100}{{\mathrm{OD}}_{\mathrm{untreated}\;\mathrm{cells}}-{\mathrm{OD}}_{\mathrm{blank}\;\mathrm{DMSO}}}$$

### 4.8. Analysis of Dox/siRNA Synergistic Effect by CompuSyn Software

The synergistic effect of Dox and Mcl-1 siRNA was determined by MTT assay using two complementary methods, i.e., a constant and a non-constant ratio. For the non-constant ratio experiment, the concentration of Mcl-1 siRNA was fixed at 0.1 µM for all treatment conditions, while the concentrations of Dox were increased as follows: 0.02, 0.04, 0.09, 0.180, 0.46, and 0.92 µM. For the constant ratio experiment, cells were treated with 5 different concentrations of Dox and Mcl-1 siRNA, while the ratio of Dox:Mcl-1 siRNA was maintained at 2.5 for all treatment conditions. The lowest concentration of Dox and Mcl-1 siRNA was 0.25 and 0.1 µM, while the highest concentration was set at 0.75 and 0.3 µM, respectively. Additionally, cells were also treated with each individual agent in a single treatment with the same concentration as in the combination experiments for control. Percent cell viability values were obtained for all conditions and incorporated into the calculation for synergism and the dose reduction effect by CompuSyn software using an algorithm as described to CI and dose reduction index (DRI) [68]. Percentages of cell viability were translated to Fa, which was obtained from (100% cell viability)/100. The actual data points, including treatment concentrations (µM) and Fa, were inputted into the software to automate combination data. CI was interpreted as the combination effect as follows: CI < 1 (synergist), CI = 1 (additive) and CI > 1 (antagonism), additive effect or antagonism. The value of DRI reflected the fold decrease in the concentration of each agent in combination when compared to the concentration of the same agent used individually to reach the same inhibitory effect. DRI = 1 (no dose reduction), DRI > 1 (favorable dose reduction), and DRI < 1 (favorable dose increase).

### 4.9. mRNA Expression Level Determination by Quantitative Real-Time PCR

The mRNA expression level of Mcl-1 was examined in MCF-7 cells treated with the combination of Mcl-1 siRNA and Dox at 0.1 and 0.25 µM/well, respectively, 48 h after siRNA transfection. mRNA was extracted from cultured cells in a 96-well plate and converted to cDNA with SuperPrep^™^ II Cell Lysis & RT Kit for qPCR (Toyobo, Japan). The obtained cDNA was analyzed by quantitative real-time PCR, using Thunderbird^™^ SYBR^®^ qPCR Mix (Toyobo, Japan), in LightCycler^®^ 480 Instrument II (Roche, Switzerland) with annealing condition at 59 °C, 30 sec. Specific primers spanning exons of Glyceraldehyde 3-phosphate dehydrogenase (GAPDH) and Mcl-1 were designed from NM_001357943.2 and NM_021960.5, respectively. GAPDH forward primer: TTTTGCGTCGCCAGCCG, GAPDH reverse primer: CGCCCAATACGACCAAATCC (product length 84 bp), Mcl-1 forward primer: GGAGACCTTACGACGGGTT, Mcl-1 reverse primer: AGTTTCCGAAGCATGCCTTG (product length 75 bp). The relative mRNA levels were computed with cycle threshold (CT) using the delta-delta CT method (△△CT method).

### 4.10. Analysis of Cellular Apoptosis by Hoechst 33342 and SYTOX™ Green Double Staining

Apoptosis and necrosis were investigated in MCF-7 cells by double staining assay. Cells were seeded in a 96-well plate and transfected with siRNA as described in the transfection section. After 48 h of treatment, cells were then stained with a mixture solution containing 5 µg/mL of Hoechst 33342 and 5 µg/mL of SYTOX™ Green (Invitrogen, Carlsbad, CA, USA) for 15 min. Within 1 h after staining, a picture of the cells was taken under an inverted fluorescence microscope (Eclipse TE 2000-U; Model: T-DH Nikon^®^, Tokyo, Japan). The apoptosis cells (bright blue fluorescence) and the death cells (green fluorescence) were quantified and presented as percentages from three different microscopic fields.
(2)$$\%\;\mathrm{apoptosis}\;\mathrm{cells}=\frac{\mathrm{bright}\;\mathrm{blue}\;\mathrm{fluorescent}\;\mathrm{cells}\times 100}{\mathrm{all}\;\mathrm{blue}\;\mathrm{fluorescent}\;\mathrm{cells}}$$
(3)$$\%\;\mathrm{death}\;\mathrm{cells}=\frac{\mathrm{green}\;\mathrm{fluorescent}\;\mathrm{cells}\times 100}{\mathrm{all}\;\mathrm{blue}\;\mathrm{fluorescent}\;\mathrm{cells}}$$

### 4.11. MCF-7 3D Spheroid Cell Culture

The spheroid culture was performed in a 96-well flat-bottomed plate coated with 50 µL/well of 1.5% agarose [69]. Agarose dissolved in purified water was sterilized by autoclave before use. For agarose layering, the agarose gel was melted at 60 °C, transferred to each well, and then left at room temperature for 30 min to cool down and solidify. Cells diluted as 1 × 10^3^ cells in 100 µL of completed medium were then seeded on top of agarose-layer. Cell settlement was aided with centrifugation at 400× *g* for 15 min using a Heraeus Multifuge 1S-R centrifuge (Thermo Electron Corporation, Langenselbold, Germany). The cells were then incubated for 72 h to form a spheroid. The cell culture medium was gently replenished with 50 µL of fresh medium every 24 h for 3 days. The diameter of the resulting spheroids was around 200–300 µm. Spheroids were then treated with designated samples. The culture medium was replenished every 24 h. Pictures of spheroid were taken every 24 h for 8 days using an inverted microscope. The radius of each spheroid was analyzed with Image J software (National Institute of Health, Bethesda, MD, USA). Data were reported as the relative volume of the spheroid, which was calculated using the following equations.
(4)$$\mathrm{Volume}\;\mathrm{of}\;\mathrm{spheroid}=\mathrm{Length}\;\mathrm{of}\;\mathrm{longest}\;\mathrm{axis}\times \frac{{\mathrm{Width}\;\mathrm{of}\;\mathrm{shortest}\;\mathrm{axis}}^{2}}{2}$$
(5)$$\mathrm{Relative}\;\mathrm{volume}\;\mathrm{of}\;\mathrm{spheroid}\;=\frac{{\mathrm{Volume}\;\mathrm{of}\;\mathrm{spheroid}\;\mathrm{at}\;\mathrm{day}}_{n}}{{\mathrm{Volume}\;\mathrm{of}\;\mathrm{spheroid}\;\mathrm{at}\;\mathrm{day}}_{0}}$$

## Figures and Tables

**Figure 1 pharmaceutics-13-00550-f001:**
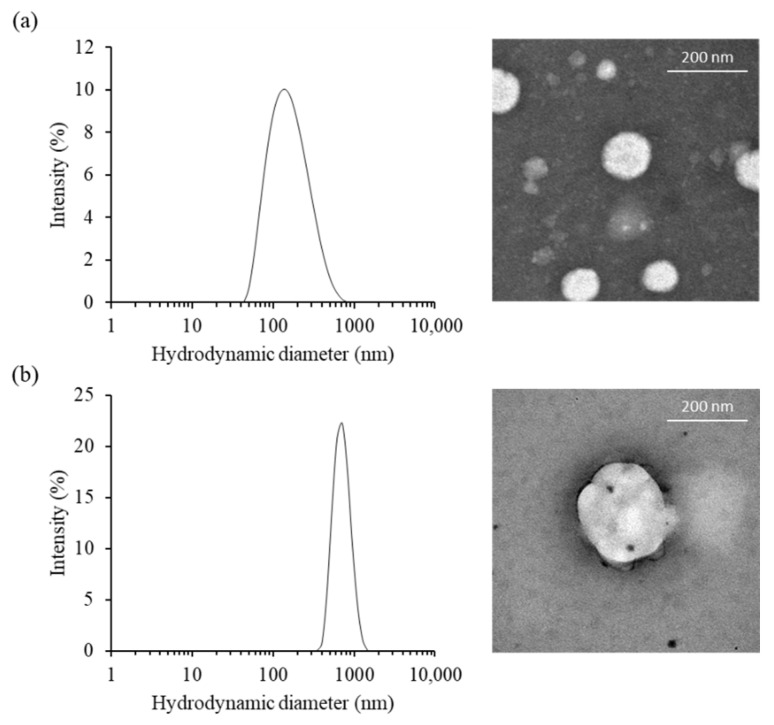
The physical characteristics of (**a**) PCN-B and (**b**) PCN-B/siRNA complexes, (**left**) the particle size distribution and (**right**) transmission electron microscopic (TEM) images.

**Figure 2 pharmaceutics-13-00550-f002:**
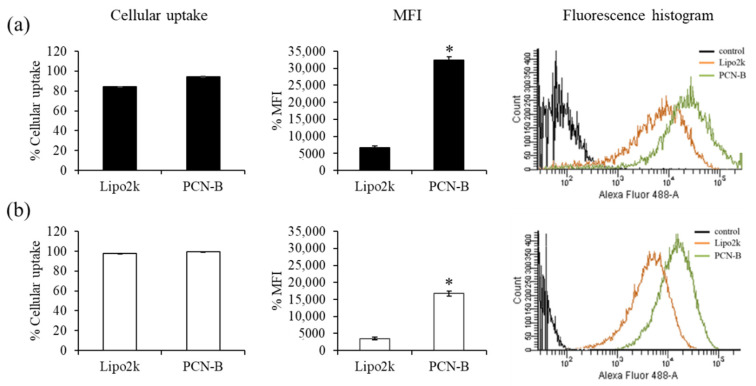
Cellular uptake study of Lipo2k or PCN-B/siAF488 complexes in (**a**) MCF-7 cells and (**b**) MDA-MB-231 cells by flow cytometry. The results were presented as the percentage of cellular uptake, mean fluorescent intensity (MFI) plot, and fluorescence histogram. * *p* < 0.05.

**Figure 3 pharmaceutics-13-00550-f003:**
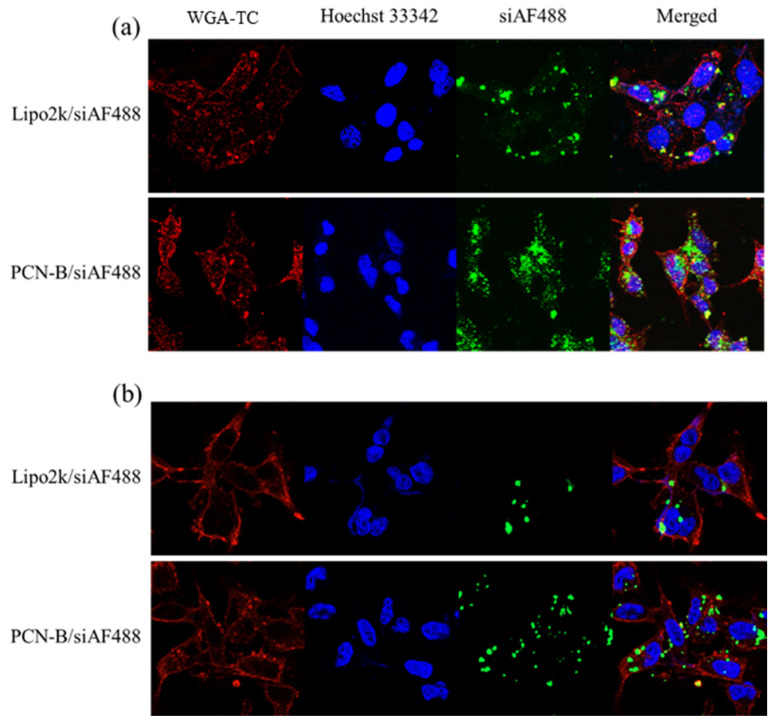
Intracellular distributions of siAF488 complexes transfected with Lipo2k and PCN-B for 24 h in (**a**) MCF-7 cells and (**b**) MDA-MB-231 cells by confocal laser scanning microscope (CLSM).

**Figure 4 pharmaceutics-13-00550-f004:**
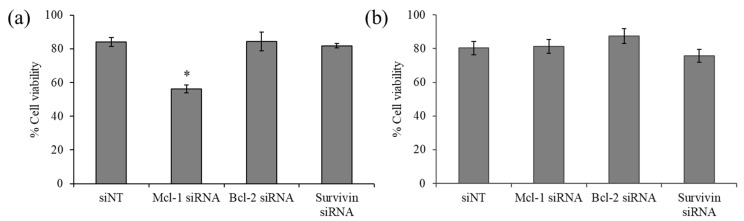
Cell viability of indicated siRNA targeting (Mcl-1, Bcl-2, and survivin) using delivery of PCN-B in (**a**) MCF-7 and (**b**) MDA-MB-231 cells. * *p* < 0.05.

**Figure 5 pharmaceutics-13-00550-f005:**
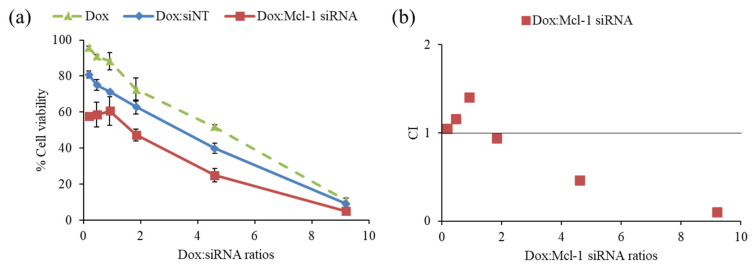
Screening synergistic effect of doxorubicin (Dox) and Mcl-1 siRNA combination at ratios of 0.2, 0.5, 0.9, 1.8, 4.6, and 9.2, (**a**) cell viability of non-constant ratio experiment and (**b**) CI-combination ratio plot. The concentration of each agent in every data point was described in the materials and methods section.

**Figure 6 pharmaceutics-13-00550-f006:**
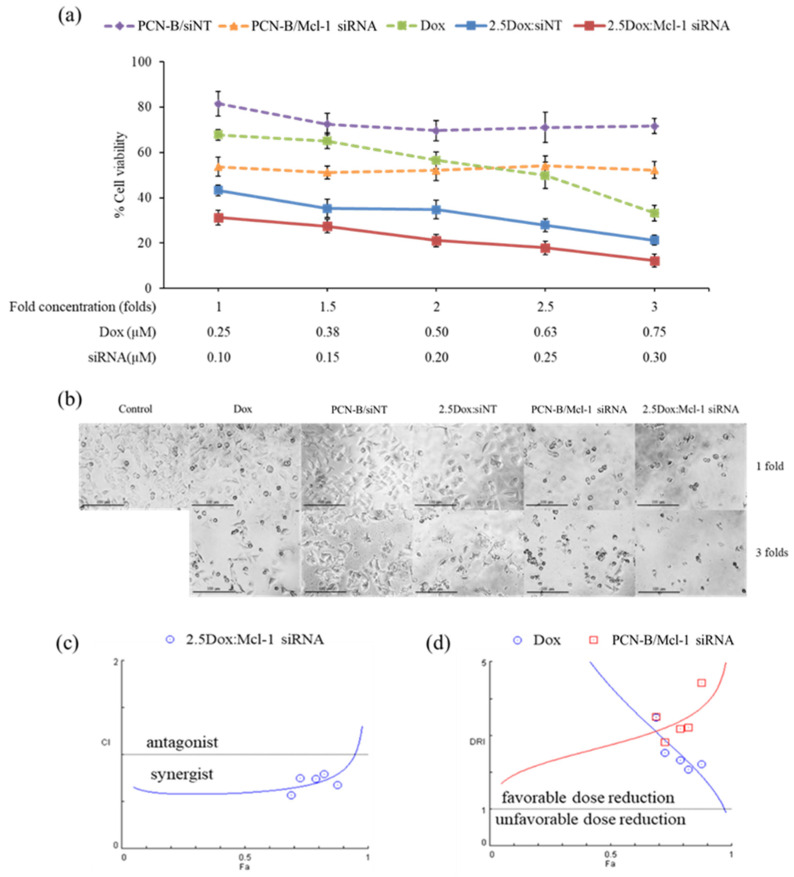
The constant ratio experiment was performed with Dox:Mcl-1 siRNA at the ratio of 2.5. The Dox and Mcl-1 siRNA concentration was simultaneously increased from 1- to 3-fold, starting with 0.25 µM of Dox and 0.1 µM of Mcl-1 siRNA. (**a**) The cell viability of MCF-7 cells treated with combination treatment and individual treatments were illustrated as solid lines and dash lines, respectively. (**b**) Morphology images after 72 h of treatments under an inverted fluorescence microscope (100X). The simulated lines were generated from CompuSyn, plotted as (**c**) CI-Fa and (**d**) DRI-Fa. The actual points were shown as circles and squares.

**Figure 7 pharmaceutics-13-00550-f007:**
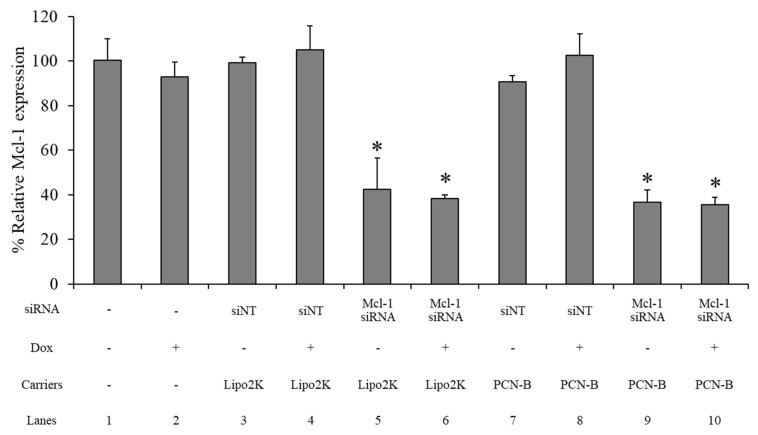
The relative mRNA expression study was evaluated using the ΔΔCT method, which presented as a percentage of the relative Mcl-1 mRNA expression in the MCF-7 cells. * The data was significantly different from the non-targeted siRNA (siNT) transfection complexes at *p*-value < 0.001.

**Figure 8 pharmaceutics-13-00550-f008:**
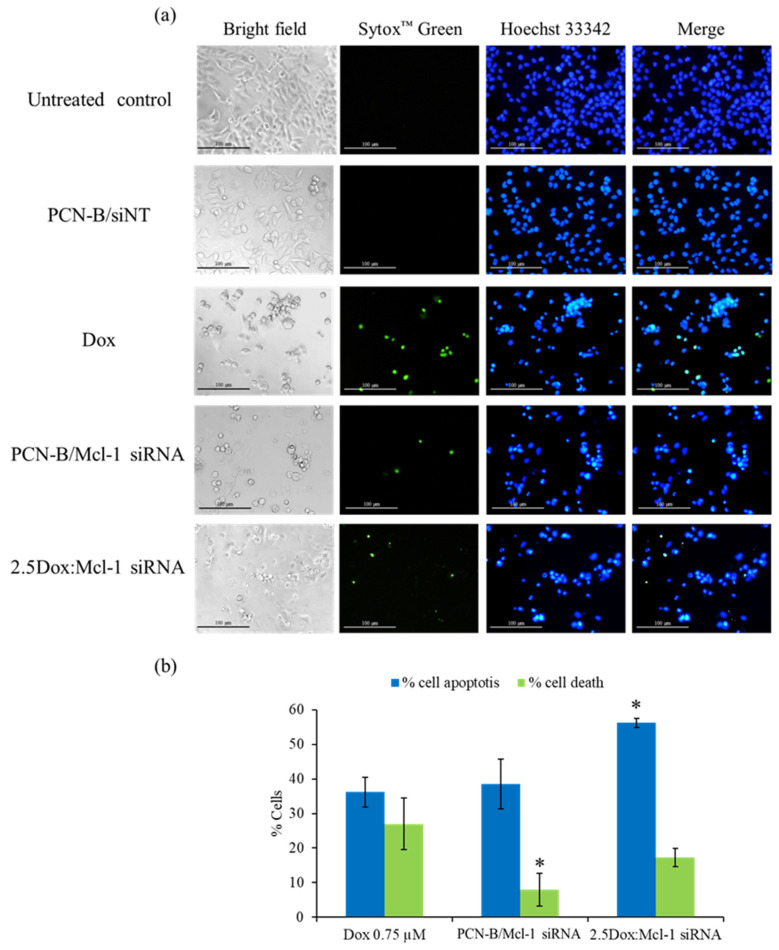
Double-stain apoptosis detection was observed under an inverted fluorescence microscope (100X) (**a**) after 24 h of treatments; untreated control, PCN-B/siNT, Dox (0.75 µM), Mcl-1 siRNA (0.1 µM), and the combination of Dox (0.25 µM) and Mcl-1 siRNA (0.1 µM) at the molar ratio of 2.5. Apoptotic and dead cells were counted and compared to all cells in the same area, presenting as % cell apoptosis (blue fluorescent cells) and cell death (green fluorescent cells) (**b**). * The data was significantly different from the Dox treatment at *p*-value < 0.05.

**Figure 9 pharmaceutics-13-00550-f009:**
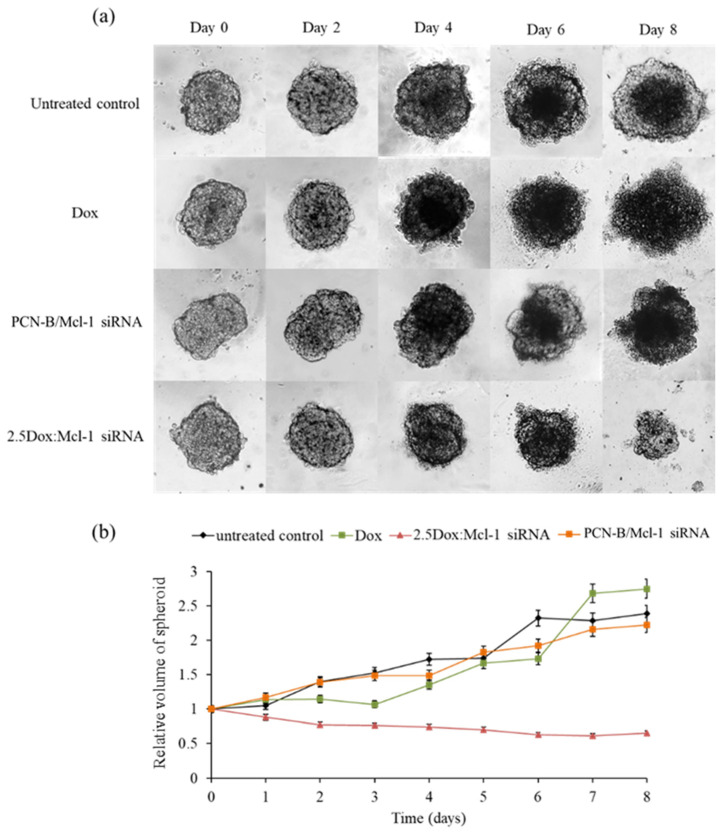
Synergistic effect on the growth of MCF-7 3D spheroid cells (**a**) spheroid images under a microscope at various time points and (**b**) relative volume of spheroids from day_0_-day_8_. The Dox and Mcl-1 siRNA concentration were used at 1.25 and 0.5 µM, respectively, for the individual treatments and the combination.

## Data Availability

The data presented in this study are available in this article: Synergistic effect of doxorubicin and siRNA-mediated silencing of Mcl-1 using cationic niosomes against 3D MCF-7 spheroids.

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
