# Peer review of "Synergistic Effect of Doxorubicin and siRNA-Mediated Silencing of Mcl-1 Using Cationic Niosomes against 3D MCF-7 Spheroids"

_pharmaceutics, 2021, doi:10.3390/pharmaceutics13040550_

Round 1

Reviewer 1 Report

This paper has an high scientific soundness and it is well written with a proper use of English.

I recommend publication after minor revisions.

Here are some issues:

Abstract. Lines 28-29. “significantly more potent”. Maybe it could be better to talk about efficiency.

The Introduction is interesting and full described in details. However, the state of the art, that is well written, should be supported by the addition of more references. For example:

-Introduction, Line 36. Please, add a reference here.

-Line 38. Even if these side effects are very well known, a reference would be appreciated.

-Line 41. Also here the addition of the information about DOX should be supported by references.

-Line 44. A double space needs to be removed before “such”

-Line 46. “higher dose”. Higher than which other dose or limit? Therapeutic window maximum concentration? Please clarify or use another expression.

Line 92. The concept of liposomes and niosomes are introduced in this sentence, but no references are present.

Line 94. Could you explain why the cost of niosomes is lower than conventional liposomes? Are the authors referring to the conventional methods of production? Or to the chemico physical characteristics of the liposomes, compared with the most recently discovered niosomes?

Preparation of niosomes.

The technique is the same conventional method for the production of liposomes, followed by sonication steps. Were there any significant changes in the production of niosomes, using this technique?

There is no explanation of the method for the measurement of the particle size of niosomes, but in paragraph 4.5 it is said that sonication is used to measure the reduced size of niosomes.

Did you determine the encapsulation efficiency of Dox into niosomes?

I suggest enlarging or dividing figures 1a-h. They are too much, and, in particular, figures 1b and 1d are embedded! It is impossible to read the embedded figures.

Do you have a particle size distribution of niosomes produced, before working with cells? Do you have a mean diameter plus/minus standard deviation?

The sentence of Line 300 starts with a reference Trabulo (2011) that is not cited according to the Journal guidelines.

Thank you

Author Response

Reviewer 1

This paper has an high scientific soundness and it is well written with a proper use of English. I recommend publication after minor revisions.

Here are some issues:

Abstract. Lines 28-30. “significantly more potent”. Maybe it could be better to talk about efficiency.

  • Thank you for your suggestion. The abstract has been rewritten in lines 28-30 as follows “Synergistic antitumor activity was further verified in a 3D spheroid culture which revealed that, in contrast to single agent treatment, the combination markedly decreased spheroid volume over time”.

The Introduction is interesting and full described in details. However, the state of the art, that is well written, should be supported by the addition of more references. For example:

-Introduction, Line 36. Please, add a reference here.

  • References has been added in line 37.

-Line 38. Even if these side effects are very well known, a reference would be appreciated.

  • A reference has been added in line 39.

-Line 41. Also here the addition of the information about DOX should be supported by references.

  • A reference has been added in line 43.

-Line 44. A double space needs to be removed before “such”

  • The double space before “such” has been removed.

-Line 46. “higher dose”. Higher than which other dose or limit? Therapeutic window maximum concentration? Please clarify or use another expression.

  • We would like to apologize for unclear explanation in this point. Per your comment, the sentence has been revised and added more information in lines 48-51 as “Evidence supported that the incidence of cardiotoxicity increases along with increasing doses. The use of Dox at 500–550, 551–600 and over 600 mg/m2 relatively increase the cardio-toxicity incidence at 4, 18 and 36%, respectively”.

Line 92. The concept of liposomes and niosomes are introduced in this sentence, but no references are present.

  • A reference has been added in line 102.

Line 94. Could you explain why the cost of niosomes is lower than conventional liposomes? Are the authors referring to the conventional methods of production? Or to the chemico physical characteristics of the liposomes, compared with the most recently discovered niosomes?

  • Per your comment, the introduction has been rewritten and explained more information in lines 102-112 as “Generally, the core component of liposomes is naturally derived phospholipids such as egg phosphatidylethanolamine to form bilayer membrane vesicles. Niosome, a new generation gene delivery system with a core component of non-ionic surfactants, is prepared through similar approaches utilized in the production of liposomes. They have emerged as safe and viable delivery alternatives to the liposomes with key attractive features including biodegradability, biocompatibility and less immunogenicity as evidenced by numerous preclinical and clinical researches. Moreover, nonionic surfactants possess higher chemical stability than phospholipids which can be degraded by oxidation or phospholipases. The use of synthetic surfactants is generally more economical and reduces batch to batch differences associated with the usage of lipid derived from natural source.”.

Preparation of niosomes.

The technique is the same conventional method for the production of liposomes, followed by sonication steps. Were there any significant changes in the production of niosomes, using this technique?

-           The preparation method of liposomes and niosomes is basically same but they are different in a core component. Therefore, we described more details into the introduction in lines 102-111.

There is no explanation of the method for the measurement of the particle size of niosomes, but in paragraph 4.5 it is said that sonication is used to measure the reduced size of niosomes.

  • Per your comment, we have added the measurement of the particle size of niosomes in the method of Cationic niosome preparation in lines 439-442.

Did you determine the encapsulation efficiency of Dox into niosomes?

  • Dox is not loaded in the niosomes. This study only uses the physical mixture between Dox and siRNA/cationic niosomes complexes.

I suggest enlarging or dividing figures 1a-h. They are too much, and, in particular, figures 1b and 1d are embedded! It is impossible to read the embedded figures.

  • Per your comment, Figure 1 has been separated as Figure 2 and 3.

Do you have a particle size distribution of niosomes produced, before working with cells? Do you have a mean diameter plus/minus standard deviation?

  • In accordance with your recommendation, the particle size distribution of niosomes and niosome/siRNA complexes were added in the results of characterization of cationic niosomes in lines 136-146 and in Figure 1.

The sentence of Line 300 starts with a reference Trabulo (2011) that is not cited according to the Journal guidelines.

  • The citation has been corrected in lines 348 as “Trabulo et al.”.

Reviewer 2 Report

The proposed study is very comprehensive and with a clear goal.

In this study, the authors investigated the role of the synergistic effect of doxorubicin and siRNA-mediated Mcl-1 attenuation using cationic niosomes against 3D MCF-7 spheroids.

The paper includes interesting information on the importance of combination therapy with chemotherapeutic agents together with gene therapy, as a potential cancer treatment strategy that improves the therapeutic efficacy of conventional therapy and reduces its negative effects.

The authors of this manuscript presented the basic mechanisms of cleon cationic niosomes B (PCN-B) for the transmission of siRNA against antiapoptotic mRNA into MCF-7 and MDA-MB-231 cells.

The special importance of this manuscript is the fact that silencing McL-1 significantly reduces the viability of MCF-7 cells and initiates apoptosis. In addition, computer modelling suggests that the combination of doxorubicin and Mcl-1 siRNA shows a synergistic relationship and allows the dose of each drug tested to be reduced by 1.71 to 3.91-fold. Also, a significant inhibitory effect of 90% was achieved, which is not the case when single-agent treatment is applied.

The applied in vitro methods used have shown that combination therapy is significantly more powerful than treatment with a single agent.

The strength of the manuscript lies in the fact that the authors confirmed that the combination therapy between the suppression of McL-1 and doxorubicin shows a synergistic effect that can be used for a new treatment for breast cancer.

In general, the manuscript is well written and the reviewer found no errors or omissions in the manuscript. All research activities were performed in detail.

Statistical processing of the obtained results and their categorization were also systematically performed.

Material and methods: Nice and clearly written.

Results: This section is very nicely written with all the necessary and concise accompanying explanations.

The results correspond to the objectives of the study. The figures are in a satisfactory resolution so that all the mentioned and explained details are clearly visible.

Discussion: The discussion part is explained to a completely satisfactory extent.

The references used are carefully selected and also up-to-date.

Author Response

Reviewer 2

Thank you for your time reading our manuscript carefully. We are very delighted with the comments and suggestions.

Reviewer 3 Report

The authors studied the Synergistic effect of doxorubicin and siRNA mediated silencing of Mcl-1 using cationic niosomes against 3D MCF-7 spheroids. The major concern needs to be addressed before the paper is accepted for publication. Followings are recommended for the manuscript.

The format of the pharmaceutics journal has not been followed in the manuscript

What is the novelty of work?

In the introduction section, the prior state of the art related to the drug delivery system is missing.

If authors have not developed niosomes, then the title of the manuscript needs to be modified.

The formulation related to niosomes are missing in the introduction

Figure 4 is not very clear.

There is no method, characterization related to niosomes are available.

The vocabularies, grammar and writing style in the manuscript must be revised and improved.

Authors should define abbreviation first time after that can use abbreviation.

Did the authors check the cytocompatibility of the combination system in normal cells line?

Author Response

Reviewer 3

The authors studied the Synergistic effect of doxorubicin and siRNA mediated silencing of Mcl-1 using cationic niosomes against 3D MCF-7 spheroids. The major concern needs to be addressed before the paper is accepted for publication. Followings are recommended for the manuscript.

The format of the pharmaceutics journal has not been followed in the manuscript

  • Per your comment, the manuscript format has been revised following to the guideline of the journal.

What is the novelty of work?

  • Per your comment, we revised the introduction Lines 123-127 as “Here, this study provided a systematic approach to identifying the synergistic effect of Dox and a potent siRNA delivered by cationic niosomes, hypothesized as a novel therapeutic strategy for breast cancer treatment. Moreover, there is still a lack of empirical algorithm study in the synergistic effect of combining chemotherapeutic agent and siRNA.”.

In the introduction section, the prior state of the art related to the drug delivery system is missing.

  • Per your comment, we have added more details in lines 93-99 as “Current siRNA delivery systems can be classified into two categories i.e. viral and non-viral vectors. Although viral-vectors provide high transfection efficiency, they also possess several drawbacks, for instance, immunogenicity and low loading capacity. Therefore, non-viral vectors have emerged as alternative delivery systems with minimal safety concerns which are broadly used for gene delivery. The advantages of non-viral nanoparticles include the simple production process and ease of structural modification.”.

  • and lines 104-116 as “Niosome, a new generation gene delivery system with a core component of non-ionic surfactants, is prepared through similar approaches utilized in the production of liposomes. They have emerged as safe and viable delivery alternatives to the liposomes with key attractive features including biodegradability, biocompatibility and less immunogenicity as evidenced by numerous preclinical and clinical researches. More-over, nonionic surfactants possess higher chemical stability than phospholipids which can be degraded by oxidation or phospholipases. The use of synthetic surfactants is generally more economical and reduces batch to batch differences associated with the usage of lipid derived from natural source. Cationic lipids are often combined in lipid vesicles, offering the adjustable electrostatic interaction with the cellular mem-brane and genetic materials to facilitate gene delivery through the selection of cationic lipids with various structure. However, there is only a few studies investigating the application of cationic niosomes for gene delivery.”.

If authors have not developed niosomes, then the title of the manuscript needs to be modified.

  • Per your comment, we added the preparation method and characterization in the method of cationic niosome preparation in lines 444-448. The particle size distribution of niosomes and niosome/siRNA complexes were added in the results of characterization of cationic niosomes in line 136-146 and in Figure 1.

The formulation related to niosomes are missing in the introduction

  • In accordance with your recommendation, we revised the introduction in lines 102-114.

Figure 4 is not very clear.

  • Per your comment, we revised Figure 4 (Figure 6 of the revised manuscript) by adding more explanation of each treatment concentration in x-axis of Figure 6a and Figure caption in lines 253-256 as “Figure 6. The constant ratio experiment was performed with Dox:Mcl-1 siRNA at the ratio of 2.5. The Dox and Mcl-1 siRNA concentration was simultaneously increased from 1 to 3 folds, starting with 0.25 µM of Dox and 0.1 µM of Mcl-1 siRNA. (a) The cell viability of MCF-7 cells treated with combination treatment and individual treatments were illustrated as solid lines dash lines, respectively.”.

There is no method, characterization related to niosomes are available.

  • Per your comment, we have added the measurement of the particle size of niosomes in the method of cationic niosome preparation in lines 444-448.

The vocabularies, grammar and writing style in the manuscript must be revised and improved.

  • The English language has been corrected and edited by an English editing service.

Authors should define abbreviation first time after that can use abbreviation.

  • Per your comment, abbreviations in this manuscript have been corrected.

Did the authors check the cytocompatibility of the combination system in normal cells line?

  • We do not check it in normal cells. We realized that the cytocompatibility of the system in normal cell lines is also beneficial knowledge.

Round 2

Reviewer 3 Report

Authors have responded correctly most of thd comment. Paper can be accepted in its current form.